# *Azospirillum brasilense* in the Planting Furrow of Sugarcane to Minimize the Use of N Fertilizer

**DOI:** 10.3390/plants14111599

**Published:** 2025-05-24

**Authors:** José Augusto Liberato de Souza, Lucas dos Santos Teixeira, Gabriela da Silva Freitas, Lucas da Silva Alves, Maurício Bruno Prado da Silva, Juliana Françoso da Silva, Fernando Shintate Galindo, Carolina dos Santos Batista Bonini, Clayton Luís Baravelli de Oliveira, Reges Heinrichs

**Affiliations:** 1Faculty of Engineering, São Paulo State University—Unesp, Ilha Solteira 15385-000, SP, Brazil; jose.augusto-liberato-souza@unesp.br (J.A.L.d.S.); gabrielafreitas1997@gmail.com (G.d.S.F.); 2Department of Crop Science, São Paulo State University—Unesp, Dracena 18610-034, SP, Brazil; santos.teixeira@unesp.br (L.d.S.T.); mauricio.prado@unesp.br (M.B.P.d.S.); jufrancoso28@gmail.com (J.F.d.S.); fernando.galindo@unesp.br (F.S.G.); carolina.bonini@unesp.br (C.d.S.B.B.); 3Faculty of Agricultural and Veterinary Sciences, São Paulo State University—Unesp, Jaboticabal 14884-900, SP, Brazil; silva.alves@unesp.br; 4Federal Institute of Education, Science and Technology of São Paulo—IFSP, São Roque 01109-010, SP, Brazil

**Keywords:** biological nitrogen fixation, growth-promoting bioinoculants, nutrient use efficiency, *Saccharum* spp.

## Abstract

Sugarcane (*Saccharum* spp.) stands out in the context of sustainable agricultural production due to its versatility and energy potential. However, management challenges, such as nitrogen (N) fertilization associated with microbiological action, require improvement. In this context, the use of the bacterium *Azospirillum brasilense* has been studied as an alternative to reducing the use of mineral fertilizers. The objective of this study was to evaluate the application of *Azospirillum brasilense* in the planting furrow of sugarcane in terms of leaf diagnosis, nutrient uptake, yield and technological quality of the stalks, and total fresh and dry biomass of the aerial parts of the plants. The experiment was conducted under field conditions at two locations during the 2022/2023 growing season. The soils in Areas 1 and 2 were classified as medium-textured oxisol and sandy-textured oxisol, respectively. The experimental design was a randomized block design with four replications. The treatments were as follows: (T1) 28 kg ha^−1^ of N; (T2) 14 kg ha^−1^ of N; (T3) T2 + 0.2 L ha^−1^ of inoculant; (T4) T2 + 0.4 L ha^−1^ of inoculant; (T5) T2 + 0.6 L ha^−1^ of inoculant; (T6) T2 + 0.8 L ha^−1^ of inoculant. In Area 1, treatment T5 showed a total fresh biomass yield of the aerial parts that was 34% higher than T2. Total dry biomass, tillering, stalk yield, and technological parameters did not differ significantly between treatments in either area. In terms of nutrient uptake, treatment T5 consistently ranked among those with the highest averages for P, K, Ca, Mg, S, Fe, Mn, and Zn in both experimental areas. The dendrogram showed similar results between treatments T1 and T5. The application of 0.6 L ha^−1^ of the solution containing *Azospirillum brasilense*, combined with 50% of the recommended N dose, increased total fresh biomass production. Total dry biomass, stalk yield, tillering, and technological variables of the crop were not affected by the presence of the bacterium.

## 1. Introduction

Brazil is the world’s leading producer of sugarcane (*Saccharum* spp.), setting a record in the 2023/2024 harvest by processing 713.2 million tons [1]. Sugarcane is one of the most important crops globally, with diverse uses, including food production (liquid, refined, crystal, and organic demerara sugars), beverages, biofuels (as a feedstock for ethanol), renewable energy through ethanol and biogas, and biomaterial production using bacterial cellulose [2,3].

Several factors can influence sugarcane production, with nutrient management and nutrient assimilation rates playing critical roles. Medium-textured soils with low organic matter content are common in tropical conditions, offering highly favorable physical properties for agricultural production. However, they present chemical limitations [4], which can be mitigated through effective management and fertilization strategies.

The nutritional requirements of sugarcane vary according to the expected yield and crop cycle. The crop’s developmental stage directly influences the highest nutrient uptake around 200 days after planting. In the soil/nutrient dynamic, various reactions can affect nutrient availability and plant absorption, directly influencing the amount of mineral fertilization required [5].

Among the most demanded nutrients by sugarcane is nitrogen (N), a vital element for metabolic processes and an essential component of amino acids, proteins, and nucleic acids, contributing directly to chlorophyll biosynthesis [6]. Urea [CO(NH_2_)_2_] is the most used source to meet plant N requirements due to its industrial advantages, such as high N concentration per unit mass (45–46%) and lower production costs compared to other N sources [7]. Once applied to the soil, urea is hydrolyzed by the enzyme urease, producing ammonia (NH_3_-N), which is rapidly lost to the atmosphere as gas [8,9].

Mineral fertilization can be employed to meet the crop’s N demand; however, this increases production costs. Moreover, the long-term intensive use of these inputs can lead to adverse environmental impacts, such as soil acidification and contamination of water resources by chemical compounds present in fertilizers, among other effects [5]. In sugarcane cultivation, fertilizers make up 20–30% of the production costs, with a rather variable range of fertilization recommendations, both for plant cane and ratoon crops. In Brazil, more than 15% of the fertilizers are used for this crop [10,11]. Several strategies have been used to increase fertilization efficiency. The most frequently used strategy has been the application of enhanced-efficiency fertilizers [12] One approach to reducing reliance on mineral N fertilizers is the use of plant growth-promoting bacteria (PGPB), which has shown promising results in agricultural production systems [13].

PGPB have the potential to stimulate plant growth through biological N-fixation. They contribute directly to greater root system development, the synthesis of phytohormones, and phosphate solubilization and may also enhance plant resistance to both biotic and abiotic stresses [14].

The N-fixing ability of these bacteria grants them diazotrophic capabilities, making their use in agriculture a highly sustainable alternative. This reduces the need for mineral fertilizers and, consequently, dependence on non-renewable industrial processes [15,16].

*Azospirillum* is a well-studied genus of bacteria due to its ability to fix atmospheric N (N_2_) and produce phytohormones that enhance plant development. Scudeletti et al. [17], when evaluating doses of *Azospirillum brasilense* in sugarcane, observed an increase in sugar production, energy output, and yield when the inoculant was applied during the sprouting and tillering stages. Additionally, Ferreira et al. [18] highlighted the potential for improving raw material quality, which can translate into strategic gains in final sugarcane productivity.

The objective of this study was to evaluate the application of *Azospirillum brasilense* in the planting furrow of sugarcane, combined with reduced N fertilization, focusing on the plant’s nutritional status, nutrient extraction from the aerial parts, stalk productivity and technological quality, and total fresh and dry biomass production.

## 2. Results

### 2.1. Effects on Sugarcane Technological Quality and Yield

Regarding sugarcane productivity and tillering, responses were observed in Area 1 for total fresh matter (FM), with treatment T5 differing from the other treatments. For total dry matter (DM), the following relationship was observed: T1 > T3, T4, and T5 > T2 and T6 (Table 1). The other treatments applied in Area 1 were not significantly affected by the nutrient sources.

The technological parameters are presented in Table 2. No significant differences were observed for any of the analyses in either area (Table 2).

### 2.2. Plant Nutritional Status

Leaf nutrient contents were affected by the treatments in both areas. In Area 1, treatments T1, T3, and T6 differed significantly from the others in terms of P concentrations. For Ca, treatments T3, T5, and T6 showed significant differences compared to the others.

Mg levels differed in T3 and T6, while S responded to the treatments in T1 and T5. For Cu, the observed trend was T1 > T4, and T5 > T6 > T3 > T2. Regarding Fe, treatments T2, T3, T4, and T5 differed significantly from T1 and T6. Mn content was significantly lower in T4 compared to the other treatments, and Zn showed a significant response in T5. N and K did not respond to the applied sources (Table 3).

In Area 2, the nutrients N, K, Ca, Mg, S, and Zn were not significantly affected by the treatments (Table 3). However, treatments T3, T5, and T6 differed significantly for P. For Cu, T1, T2, and T3 were significantly different from the other treatments. For Fe, T1, T3, and T5 differed from the others, while for Mn, T1 and T3 showed significant differences compared to the remaining treatments (Table 3).

For stalk nutrient concentrations, no significant responses were observed only for N and Cu in Area 2 (Table 4). The macronutrient levels responded as follows: For N, T5 > T1, T3, and T6 > T2 and T4; for P, T5 > T4 > T1, T2, T3, and T6; for K, T5 > T2, T4, and T6 > T1 and T3; for Ca, T5 > T1 > T2, T3, T4, and T6; for Mg, T5 differed from the other treatments; and for S, T2 showed the lowest value among treatments.

Regarding micronutrients, for Cu, T1 and T5 differed from the other treatments. For Fe, T2 was statistically lower than the others. For Mn, T5 > T1, T3, T4, and T6 > T2, and for Zn, T1 and T5 differed from the remaining treatments (Table 4).

### 2.3. Weighted Correlation Network

For the multivariate analysis, the division and characteristics of the areas were not considered, focusing solely on the variables in question and their relationships with the applied treatments. The results are presented in Figure 1 and Figure 2.

Observing the correlation network in Figure 1, some expected relationships were confirmed. For example, the positive correlation among TRS kg ha^−1^, Brix, and Pol shows strong positive correlations: TRS kg ha^−1^—Brix, TRS kg ha^−1^—Pol, and Brix—Pol (Figure 1). Sulfur extraction also has a high positive correlation with leaf S content as well as leaf N and K content. These correlations suggest that improvements in one variable can positively influence the others, thereby enhancing production efficiency.

Fiber was negatively correlated with tilling at 90 DAT (T90), leaf Cu, and Mg content. N extraction was positive correlated with FPT and with Cu and Zn extraction. Furthermore, a synergistic effect was observed between nutrient extraction and Ca, Cu, Mn, Zn, and Mg (Figure 1).

In the dendrogram analysis, two large clusters were observed. One cluster grouped with T2 (14 kg of N ha^−1^), along with the respective treatments T4 and T6. The other cluster grouped with T1 (28 kg of N ha^−1^), along with the respective treatments T5 and T3. It was observed that, for the first cluster, treatments T4 and T6 are like T2 because they do not show significant values for the extraction of Mn, Cu, Ca, Zn, and Fe and the leaf content of Fe and Mn.

Conversely, the second cluster (grouped with T1) showed higher values for the extraction of Mn, Cu, Ca, Zn, and Fe and the leaf content of Fe and Mn, indicating a positive correlation of these factors in these treatments. In terms of productive parameters (TPDM), this cluster showed a greater significance compared to the first cluster. Upon closer examination of the cluster with the best agronomic performance, it was noted that T5 stood out from T1 and T3 at a second hierarchical level. T5 exhibited higher values (represented by the warmer colors observed) for the extraction of N, P, K, and FTPTP and Fe and Zn content, indicating that T5 was the best agronomic result in this study.

## 3. Discussion

A significant difference was observed in total fresh and dry matter production (Area 1). In the total aboveground fresh mass production, the application of 0.6 L ha^−1^ of the inoculant solution resulted in the highest yield, reaching 216.33 Mg ha^−1^. These results are consistent with those found in the literature, which highlights the use of PGPB-based inoculants as an effective practice to increase productivity in crops such as sugarcane [19].

The fresh mass production in treatment T5 was 34% higher than in treatment T2. *Azospirillum* can stimulate plant growth through mechanisms such as biological N-fixation, phytohormone production, and phosphate solubilization [20,21]. For total aboveground dry mass production, the highest yield was observed in T1. These results are like those reported by Fukami et al. [22], who noted that excessive doses of inoculants can have antagonistic effects due to an imbalance in soil microbiota or the increase in inhibitory substances, leading to negative impacts on production. Stalk productivity in both experimental areas exceeded the regional average, which, in the 2022/2023 season, was 80.47 Mg ha^−1^ [1]. Sugarcane plants tend to exhibit the highest productivity in the first harvest, and it tends to decline in subsequent seasons [23].

Table 2 presents the results of the technological analyses of sugarcane in two experimental areas. The technological parameters (Table 2) did not show a response to the treatments. These findings are supported by Scudeletti et al. [17], who, when evaluating the application of *Azospirillum brasilense* in plant cane and first ratoon, did not observe a significant increase in the crop’s technological parameters. However, the data related to the technological parameters are in line with those recommended by [24].

The responses observed for P in both areas can be attributed to *Azospirillum*’s ability to promote root growth, which increases the soil exploration area and consequently enhances phosphorus uptake, a nutrient that is relatively immobile in the soil [25]. Additionally, *Azospirillum* can release organic acids and phosphatases that solubilize soil P, making it more available to the plant [26]. Some authors report that *Azospirillum* may act synergistically with phosphate fertilizers, enhancing P use efficiency, which is crucial for crops such as sugarcane, with high P demand [21].

In Area 1, the concentration of Cu in the leaves observed in T1 was significantly higher than in all other treatments, while in Area 2, T1 was statistically like T2 and T3. The behavior observed for Cu can be explained by the possible saturation of absorption sites or changes in Cu availability in the soil due to microbial activity induced by *Azospirillum brasilense*. This microorganism can influence Cu availability to a lesser extent than mineral N, directly impacting the absorption of various micronutrients [27]. The decrease in Cu concentration at higher doses may indicate that an increase in the microbial population could have led to Cu immobilization in the soil or competition between microorganisms and plants for the element, reducing its availability to plants [25]. Moreover, Cu is an essential micronutrient for many enzymatic functions, but its absorption can be limited by the presence of high levels of P or other antagonistic elements [28].

Fe showed a significant difference in both areas studied. In Area 1, treatments T2, T3, T4, and T5 had higher concentrations compared to T1 and T6, whereas in Area 2, the highest concentrations were observed in treatments T1, T3, and T5, which differed statistically from the other treatments. It was noted that T6 had lower responses compared to the other treatments in both areas, indicating that high doses of the inoculant may reduce the concentration of this element in sugarcane leaves. This pattern may be related to the excessive production of siderophores by *Azospirillum*, which, when overproduced, can limit the nutrient’s availability to plants [20].

Mn responded significantly to the inoculant application in both areas. In Area 1, there were no differences between treatments, while in Area 2, treatments T1 and T3 were statistically different from the others. For Zn, the only significant response was observed in T5 in Area 1.

The foliar concentrations of Fe and Zn were within the adequate range for sugarcane, which is Fe (40.0–250.0 mg kg^−1^) and Zn (10.0–50.0 mg kg^−1^). For Mn, the concentrations exceeded the adequate range, which is 10.0–50.0 mg kg^−1^, while Cu was below the adequate range in all cases (6.0–15.0 mg kg^−1^) [29].

Table 4 shows the average extraction of macro- and micronutrients from the aboveground part of sugarcane in two experimental areas. The extraction of elements N, P, K, Ca, Mg, S, Cu, Fe, Mn, and Zn in Area 1 responded to the combined application of microorganisms and half of the mineral N fertilization. The same response was observed in Area 2, except for N and Cu (Table 4).

For N, no significant difference was observed between treatments. However, T5 significantly contributed to the extraction of P, K, Ca, and Mg, with this treatment showing higher averages than T1 in both areas for P extraction. This demonstrates the potential of *Azospirillum* to enhance fertilizer use efficiency. This combination suggests that *Azospirillum* can improve P availability and absorption, possibly by promoting root growth, better soil exploration, and increased solubilization [30,31]. Research by Hungria et al. [21] indicated that the use of *Azospirillum* in crops such as corn and sugarcane can improve the efficiency of N and P extraction, reducing fertilizer costs and environmental impacts.

In Area 1, the highest extraction of K, Ca, and Mg occurred with the application of T5, which differed significantly from the other treatments according to the Scott–Knott test. In Area 2, no significant differences were observed among treatments for K, but for Ca and Mg, the highest extractions were in T1, T3, T5, and T6 (Table 4). The greater extraction of K in Area 1 may be related to *Azospirillum brasilense*’s ability to promote root growth and increase the absorption surface, which is particularly important for K uptake. K is a mobile nutrient in the soil and is essential for various physiological functions, including osmotic regulation and enzymatic activation [32]. No significant differences were observed among treatments for S extraction in either area.

Although the observed responses for Ca, Mg, and S are consistent with the trends in concentrations, the microorganism’s ability to sustain these extraction levels suggests that the inoculant could be used as a tool to stabilize plant nutrition, even in scenarios where soil nutrient availability is limiting [33].

The extraction of all micronutrients analyzed (Cu, Fe, Mn, and Zn) in Area 1 responded significantly to the application of the inoculant solution, whereas in Area 2, only Cu showed no significant response. In Area 1, the extraction of Cu and Zn showed similar behaviors, with T5 being the only treatment to respond equally to T1 (control). In Area 2, for Zn extraction, T4 was the only treatment that responded differently and inferiorly to the others.

For Mn extraction in Area 1, treatment T5 was statistically superior to all other treatments, while in Area 2, T3 responded statistically the same as T1, with these two treatments showing the highest extractions. In terms of Fe extraction, no significant differences were found among treatments in Area 1; however, in Area 2, the lowest extractions were observed in T2 and T4. The results for Fe in both areas suggest that although *Azospirillum* can enhance Fe extraction, its effectiveness may be limited compared to traditional mineral fertilizers. This limitation may be related to how Fe is available in the soil and the plant’s ability to translocate it after absorption [27].

The results for Fe and Zn are particularly relevant, as these micronutrients play crucial roles in biochemical and physiological processes, such as photosynthesis and chlorophyll synthesis, which are vital for the healthy growth of sugarcane [34].

Stalk yield and TRS t ha^−1^ were also strongly linked, with a positive correlation of 0.86, indicating that the amount of biomass is directly related to TRS t ha^−1^ (Figure 2). This synergy can be maximized through proper nutrient management and agricultural practices that promote plant development [35].

Negative relationships were observed for cane fiber (Fbr), which negatively impacted the accumulation of Cu (CuF) and Mn (MnF) in the leaves, with correlation values of −0.96 and −0.91, respectively, and for stalk tillering at 90 days (P90), with a correlation of −0.90. In both cases, these are strong negative correlations (Figure 2). This suggests that an increase in fiber content may be associated with a reduced availability or absorption of these micronutrients, which could affect plant development. However, excessive fiber is a limiting factor for the quality of raw materials for industrial purposes [36].

Regarding N, the relationship between N in the leaves (NF) and K in the leaves (KF) was strongly correlated (0.88). The extraction of N (ExN) was also strongly associated with the extraction of various micronutrients (Figure 2), highlighting a beneficial relationship consistent with nitrogen’s role in promoting the absorption and mobilization of other essential nutrients.

The resulting dendrogram, presented in Figure 3, illustrates the hierarchical structure of the clusters. The observations were grouped into two clusters, defined according to the cut-off point established by the analysis. This cut-off point was determined based on the visual inspection of the dendrogram and the analysis of the consistency of the groupings. The treatments T2 and T4 exhibited the most consistent patterns, with a separate grouping for T5, T1, and T3.

Notably, the proximity between T1 and T5 formed a superior cluster compared to the others. This result suggests that the application of the inoculant creates conditions very similar to those achieved with conventional mineral fertilizers, indicating that inoculants may be as effective as fertilizers.

## 4. Materials and Methods

### 4.1. Experimental Site and Treatments

The experiment was conducted in two sugarcane experimental areas in the western region of São Paulo State, during the 2022/2023 growing season. The soil in Area 1 was previously classified as LATOSSOLO VERMELHO-AMARELO, with a sandy texture, and in Area 2 as LATOSSOLO AMARELO [37], with a medium texture, both corresponding to an oxisol [38]. The physical and chemical attributes are described in Table 5 [39,40]. Microbial population counts were conducted, considering diazotrophic microorganisms present in the soil (Table 6). The climate of both areas was classified as Aw according to Köppen [41] (Figure 3).

The experiment was established between April and June 2022. The experimental design in both areas was arranged in randomized blocks with four replications. The sugarcane varieties studied were RB 97-5242 (Area 1) and CTC 4 (Area 2). The experimental plots measured 90 m^2^, consisting of six sugarcane rows, spaced 1.5 m apart with a row length of 10 m. A 2.0 m gap was left between plots, and the distance between blocks was equivalent to two rows, which was used as a passageway.

The treatments consisted of five doses of an inoculant solution containing *Azospirillum brasilense* strains Ab-V5 and Ab-V6at a concentration of 2 × 10^8^ CFU mL^−1^, combined with 14 kg ha^−1^ of mineral N, and a control treatment (T1) with 28 kg ha^−1^ of mineral N. The sources of N, P_2_O_5_, and K_2_O used in treatments T2, T3, T4, T5, and T6 were NPK 05-25-25 (275 kg ha^−1^) and Hiphós 28 (357 kg ha^−1^). In T1, the same sources and doses were used, with the addition of urea (31 kg ha^−1^) to supplement N. Mineral fertilizers were applied manually at the time of sugarcane planting. Immediately afterward, the solution containing the microorganism was manually applied over the sugarcane stalks, which had already been placed in the open planting furrows. This application was carried out using a graduated plastic wash bottle, and the furrows were subsequently covered with soil. At 120 days after planting, 114.58 kg ha^−1^ of KCl was applied to all experimental units. These doses were adapted according to recommended standards for sugarcane planting. The data are presented in Table 7:

The mineral fertilizers were applied mechanically at the time of sugarcane planting. Immediately afterward, the solution containing the microorganism was manually applied over the open planting furrows using a wash bottle.

### 4.2. Number and Production of Stalks

The stalk count consecutively cut stalks in the four central rows, totaling 40 stalks per plot, excluding 2 m of border rows, and converting the values to TSH.

### 4.3. Sugarcane Technological Quality

Twelve sugarcane stalks per plot were selected, with the tops and dry leaves removed. The samples were sent to the sugar-energy units laboratory for technological analysis. The following parameters were measured: brix cane (Brx); pol cane; purity cane (Prz); fiber cane, and total recoverable sugar (TRS) [42]. The parameters were determined as follows:

Brix (Bj): The brix (soluble solids content, by weight, of juice) was determined using an automatic digital refractometer.

Fiber (F): Cane fiber was calculated using the following equation:F = 0.08 × WWB + 0.876,(1)
where WWB = wet weight of press bagasse.

Moisture % (M): Moisture was calculated using the following equation:M = (Wmw − Dmw)/Wmw,(2)
where Wmw = wet mass weight; Dmw = dry mass weight.

Pol of juice (S): Pol of juice (apparent sucrose content, by weight, of juice) was determined using an automatic digital saccharimeter and calculated using the following equation:S = LPol × (0.26047 − 0.00009882 × Bj),(3)
where LPol = saccharimetric reading of clarified juice; Bj = brix of the juice.

Pol of cane (PC): Pol of cane was calculated using the following equation:PC = × (1 − 0.01 × F) × C(4)
where S = pol in juice; F = fiber; C = coefficient used to convert the pol of extracted juice (S) to pol of cane (PC).

Juice purity (Q): Apparent juice purity (Q), defined as the percentage of pol about brix, was calculated using the equation:Q = 100 × S/Bj(5)
where S = pol in juice; Bj = brix in juice.

Reducing sugars in juice % (RS): The reducing sugars content (RS), by weight, of juice was calculated using the following equation:RS = 3.641 − 0.0343 × Q(6)
where Q = purity in juice.

Total recoverable sugar (TRS): Once the pol of cane (PC) and reducing sugars in cane (RSC) were obtained, TRS was calculated using the following equation:TRS = 9.526 × PC + 9.05 × RS(7)

### 4.4. Nutrition Status

The nutritional status of the plants was assessed through leaf analysis and whole-plant sampling (leaves, stalks, and tassels) for macronutrients (N, P, K, Ca, Mg, and S) and micronutrients (Cu, Fe, Mn, and Zn). For the foliar diagnosis, 20 “+1” leaves were collected per plot, using the central 30 cm of the leaf blade without the midrib. This diagnostic leaf corresponds to the first fully developed leaf located below a reference point on the plant. In the case of sugarcane, for example, it is the first fully expanded leaf with a well-defined blade tip [43].

For the whole-plant analysis, six plants were collected per plot. These samples were pre-harvested, weighed, and shredded using a *Nogueira*-type forage chopper. The material was then oven-dried at 65 °C under forced ventilation until constant weight. Dried subsamples of the whole plants were ground using a Wiley-type mill, weighed, and prepared for digestion.

N content in both leaf and whole-plant samples was determined via sulfuric acid digestion followed by titration using the micro-Kjeldahl method. Ps, K, Ca, Mg, and S contents were determined through nitric-perchloric acid digestion. P and S concentrations were measured using a light spectrophotometer, whereas K, Ca, Mg, and micronutrient concentrations were quantified using an atomic absorption spectrophotometer [43].

### 4.5. Statistical Analysis

The data were first tested for normality using the Shapiro–Wilk test. Following this, an analysis of variance (ANOVA) was performed. When significant differences were detected, means were compared using the Scott–Knott test at significance levels of *p* < 0.01, *p* < 0.05, and *p* < 0.1.

Multivariate analyses were conducted using R software (version 4.3.2). A correlation network was generated using the “qgraph” function from the “qgraph” package, and cluster analysis (Ward’s method) was performed using the “factor extra” and “ggplot” packages.

## 5. Conclusions

The application of 0.6 L ha^−1^ of the solution containing *Azospirillum brasilense* at the planting furrow of sugarcane, combined with 50% of the nitrogen dose, increased the total fresh biomass production in Area 1. However, the total dry mass, stalk productivity, tillering, and technological parameters of the crop were not affected by the presence of the bacteria.

The nutrient concentrations in the foliar diagnosis did not differ with the application of *Azospirillum* and full or half doses of nitrogen fertilization.

In nutrient extraction, the treatment with 0.6 L ha^−1^ of the inoculant solution, combined with 50% of the nitrogen dose, consistently ranked among the treatments with the highest averages of P, K, Ca, Mg, S, Fe, Mn, and Zn in both experimental areas.

According to the dendrogram, the control treatment and T5—the application of 0.6 L ha^−1^ of the inoculant solution and a half dose of mineral nitrogen—displayed a superior clustering compared to the others, with a similar coloration, indicating very comparable conditions between the two treatments.

## Figures and Tables

**Figure 1 plants-14-01599-f001:**
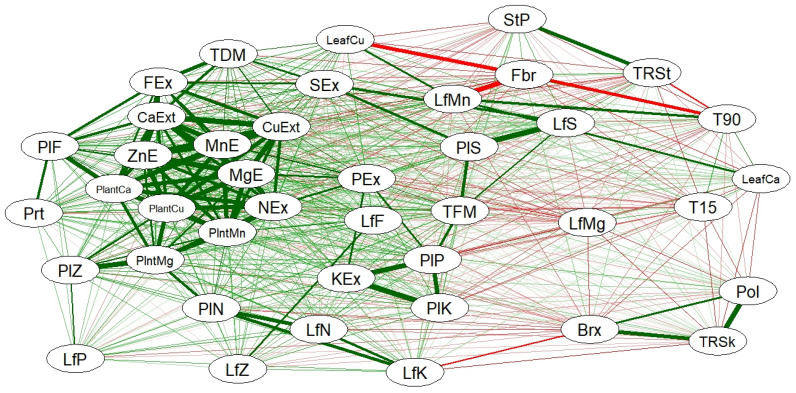
Weighted correlation network created using the statistical program R. Red connections represent negative relationships, while green connections indicate positive relationships. The varying thicknesses of the connections reflect the intensity of the relationship between the variable.

**Figure 2 plants-14-01599-f002:**
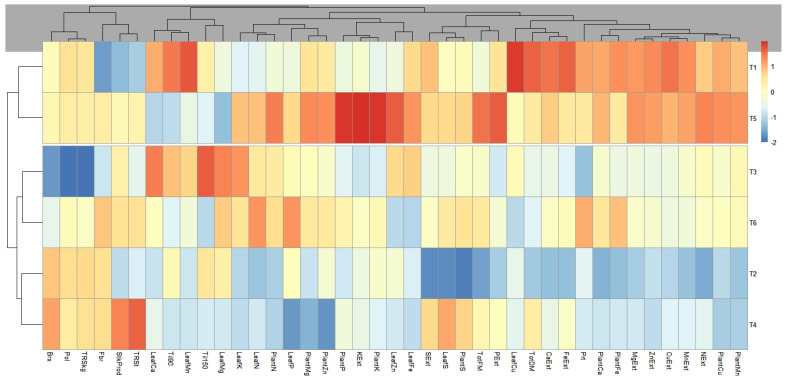
Hierarchical representation of clusters (dendrogram), created using the statistical program R. The proximity between points indicates similarity, while the branches represent the formation of clusters or groups, with the height of the connections indicating the distance between the grouped elements. Warmer colors indicate a greater influence of the respective treatment on the analyzed variable. (T1) 28 kg ha^−1^ of N; (T2) 14 kg ha^−1^ of N; (T3) T2 + 0.2 L ha^−1^ of inoculant; (T4) T2 + 0.4 L ha^−1^ of inoculant; (T5) T2 + 0.6 L ha^−1^ of inoculant; and (T6) T2 + 0.8 L ha^−1^ of inoculant.

**Figure 3 plants-14-01599-f003:**
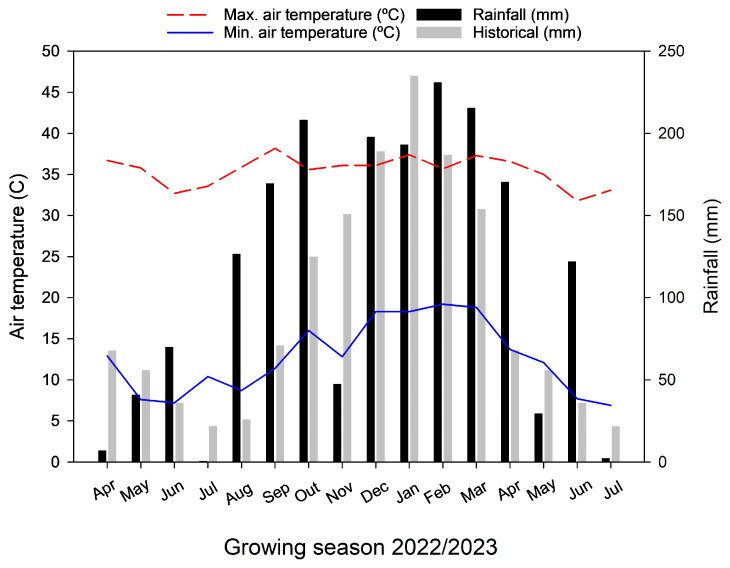
Historical and experimental period precipitation, and minimum, and maximum temperatures (April 2022 to July 2023).

**Table 1 plants-14-01599-t001:** Production and tillering of sugarcane in two experimental areas. Agricultural year 2022/2023.

Treat.	Total FM	Total DM	Tillers 90	Tillers 150	Cane Yield
	kg ha^−1^		N° m^−1^		TSH
Area 1					
T1	190.67 b	63.07 a	22.44	13.69	150.15
T2	160.42 b	39.84 c	21.31	13.52	151.67
T3	185.17 b	49.73 b	21.86	14.12	161.06
T4	186.40 b	49.70 b	21.29	12.90	171.00
T5	216.33 a	52.20 b	19.69	13.06	156.29
T6	177.42 b	43.65 c	20.81	13.81	157.21
Pr > F	°	**	ns	ns	ns
Mean	186.04	49.70	21.23	13.52	157.90
CV %	10.31	5.45	6.24	6.42	6.80
Area 2					
T1	171.72	40.52	26.52	20.04	158.96
T2	175.27	38.20	25.04	20.46	163.34
T3	180.99	40.69	26.02	21.29	172.92
T4	186.75	42.45	23.19	19.54	178.96
T5	184.67	40.21	24.60	19.46	175.21
T6	201.43	39.06	23.96	18.35	180.83
Pr > F	ns	ns	ns	ns	ns
Mean	183.47	40.19	24.89	19.86	171.70
CV %	8.38	8.21	9.22	7.86	11.75

Different letters indicate significance between treatments. (Scott–Knott). **, °, and ns indicate *p* > 0.05, *p* < 0.01, and not significant, respectively. TSH—tons of stalks per hectare. (T1) 28 kg ha^−1^ of N; (T2) 14 kg ha^−1^ of N; (T3) T2 + 0.2 L ha^−1^ of inoculant; (T4) T2 + 0.4 L ha^−1^ of inoculant; (T5) T2 + 0.6 L ha^−1^ of inoculant; (T6) T2 + 0.8 L ha^−1^ of inoculant.

**Table 2 plants-14-01599-t002:** Technological analysis of sugarcane stalks in two experimental areas. Agricultural year 2022/2023.

Treat.	Brix	Pol	Purity	Fibre	TRS	TRS
	°	%			kg t^−1^	t ha^−1^
Area 1						
T1	14.53	14.57	86.22	11.11	126.06	18.93
T2	14.61	14.28	83.79	11.24	124.00	18.74
T3	13.62	13.15	83.18	10.89	115.47	18.55
T4	14.62	14.10	83.47	10.72	123.66	21.15
T5	14.21	14.16	85.65	11.15	122.71	19.15
T6	14.37	14.36	85.87	11.07	124.06	19.66
Pr > Fc	ns	ns	ns	ns	ns	ns
Mean	14.32	14.10	84.70	11.03	122.72	19.36
VC %	4.71	6.72	2.45	3.38	5.71	9.75
Area 2						
T1	18.32	13.44	86.38	11.26	135.01	21.48
T2	18.78	13.77	85.93	11.54	138.05	22.55
T3	18.10	13.18	85.42	11.62	132.43	23.08
T4	18.94	13.77	85.97	12.06	137.99	24.67
T5	18.71	13.71	85.83	11.58	137.48	23.99
T6	18.12	13.34	86.58	11.75	133.67	24.19
Pr > Fc	ns	ns	ns	ns	ns	ns
Mean	18.49	13.53	86.02	11.64	135.77	23.33
VC %	6.70	6.75	1.83	5.56	6.28	14.41

Means comparison by Scott–Knott; ns indicates Pr > Fc not significant. Brix = concentration of soluble solids; Pol = percentage of sucrose present; TRS = amount of sugar that can be effectively extracted from sugarcane. (T1) 28 kg ha^−1^ of N; (T2) 14 kg ha^−1^ of N; (T3) T2 + 0.2 L ha^−1^ of inoculant; (T4) T2 + 0.4 L ha^−1^ of inoculant; (T5) T2 + 0.6 L ha^−1^ of inoculant; (T6) T2 + 0.8 L ha^−1^ of inoculant. ns indicates not significant.

**Table 3 plants-14-01599-t003:** Concentration of macro and micronutrients in sugarcane leaf diagnosis in two experimental areas. Agricultural year 2022/23.

Treat.	N	P	K	Ca	Mg	S	Cu	Fe	Mn	Zn
	g kg^−1^						mg kg^−1^			
Area 1										
T1	15.61	1.64 a	7.97	4.57 b	1.62 b	1.44 a	5.72 a	77.02 b	67.32 a	16.71 b
T2	15.61	1.64 a	8.33	4.35 b	1.72 b	1.35 b	1.29 e	82.21 a	62.11 a	19.15 b
T3	16.61	1.52 b	8.48	4.92 a	1.83 a	1.39 b	2.01 d	84.92 a	63.26 a	19.75 b
T4	16.05	1.48 b	8.19	4.35 b	1.77 b	1.20 b	3.43 b	81.38 a	55.19 b	15.50 b
T5	16.64	1.54 b	8.73	4.73 a	1.63 b	1.42 a	3.85 b	85.65 a	63.73 a	26.76 a
T6	16.48	1.74 a	8.43	4.93 a	1.87 a	1.29 b	2.77 c	73.05 b	62.48 a	17.35 b
Pr > Fc	ns	°	ns	°	**	*	**	°	*	**
Mean	16.17	1.59	8.36	4.64	1.74	1.35	3.18	80.82	62.35	19.20
VC %	5.18	7.53	4.46	7.48	4.87	7.52	9.44	7.73	7.24	11.65
Area 2										
T1	15.36	1.78 b	6.47	5.09	2.60	1.05	4.18 a	108.29 a	98.95 a	16.92
T2	15.12	1.84 b	5.86	4.51	2.63	0.97	4.59 a	84.88 b	61.35 b	14.97
T3	14.75	1.98 a	7.24	4.88	2.71	1,07	4.88 a	101.85 a	86.76 a	18.60
T4	14.83	1.70 b	6.06	4.65	2.38	1.40	2.38 b	82.71 b	67.86 b	16.05
T5	14.88	2.06 a	6.74	4.06	2.43	1.14	3.04 b	107.70 a	65.93 b	16.00
T6	15.18	1.96 a	6.82	4.28	2.56	1.24	2.30 b	89.68 b	69.86 b	13.24
Pr > Fc	ns	°	ns	ns	ns	ns	**	**	**	ns
Mean	15.02	1.89	6.53	4.58	2.55	1.15	3.56	95.85	75.12	15.96
VC %	6.15	9.50	12.61	21.13	8.26	23.49	15.74	9.98	13.67	18.35

Different letters indicate significance between treatments. (Scott–Knott). **, *, °, and ns indicate *p* < 0.01, *p* < 0.05, *p* < 0.1, and not significant, respectively. (T1) 28 kg ha^−1^ of N; (T2) 14 kg ha^−1^ of N; (T3) T2 + 0.2 L ha^−1^ of inoculant; (T4) T2 + 0.4 L ha^−1^ of inoculant; (T5) T2 + 0.6 L ha^−1^ of inoculant; (T6) T2 + 0.8 L ha^−1^ of inoculant.

**Table 4 plants-14-01599-t004:** Extraction of macro and micronutrients by sugarcane in two experimental areas. Agricultural year 2022/2023.

Treat.	N	P	K	Ca	Mg	S	Cu	Fe	Mn	Zn
	kg ha^−1^						g ha^−1^			
Area 1										
T1	290.25 b	52.87 c	108.22 c	113.48 b	57.59 b	41.30 a	227.19 a	4412.04 a	5029.36 b	759.27 a
T2	222.98 c	39.03 c	133.11 b	58.86 c	40.13 b	24.99 b	114.00 b	3291.48 b	2719.98 c	485.01 b
T3	351.78 b	44.60 c	80.49 c	74.67 c	48.37 b	37.08 a	143.66 b	3720.77 a	4322.43 b	649.73 b
T4	269.59 c	59.79 b	137.32 b	83.16 c	46.29 b	43.30 a	140.82 b	4583.85 a	4009.41 b	651.68 b
T5	425.68 a	73.41 a	336.19 a	119.36 a	79.37 a	44.87 a	256.04 a	4404.02 a	6861.64 a	932.90 a
T6	304.19 b	51.33 c	124.95 b	80.35 c	46.45 b	34.53 a	126.01 b	3942.82 a	4034.48 b	611.39 b
Pr > Fc	**	**	**	**	**	**	**	**	**	**
Média	310.74	53.34	153.38	88.31	53.03	37.68	167.95	4059.16	4496.22	681.66
CV %	21.22	11.34	8.88	13.72	16.57	11.11	15.66	10.82	11.59	13.34
Area 2										
T1	213.97	47.59 b	139.67 a	70.27 a	48.00 a	32.34 a	143.06	5543.46 a	2931.63 a	491.55 a
T2	175.99	44.07 b	119.51 a	41.70 b	35.37 b	21.91 b	115.27	3786.44 b	1846.33 b	511.317 a
T3	202.35	52.66 b	145.34 a	72.75 a	49.96 a	34.61 a	160.93	4990.86 a	2727.80 a	520.05 a
T4	201.00	46.31 b	106.32 a	51.39 b	35.87 b	38.68 a	117.05	4148.90 b	1666.84 b	335.96 b
T5	202.82	63.27 a	194.97 a	68.04 a	42.87 a	36.83 a	140.10	5122.23 a	2176.21 b	492.48 a
T6	209.23	49.29 b	165.48 a	70.96 a	48.82 a	36.80 a	157.69	5029.46 a	2309.04 b	531.60 a
Pr > Fc	ns	°	**	*	*	*	ns	**	*	*
Mean	200.89	50.53	145.22	62.52	43.48	33.53	139.02	4770.22	2276.31	480.49
VC %	20.15	16.93	15.70	23.80	15.57	21.06	23.87	11.70	23.56	16.05

Different letters indicate significance between treatments. (Scott–Knott). **, *, °, and ns indicate *p* < 0.01, *p* < 0.05, *p* < 0.1, and not significant, respectively. (T1) 28 kg ha^−1^ of N; (T2) 14 kg ha^−1^ of N; (T3) T2 + 0.2 L ha^−1^ of inoculant; (T4) T2 + 0.4 L ha^−1^ of inoculant; (T5) T2 + 0.6 L ha^−1^ of inoculant; (T6) T2 + 0.8 L ha^−1^ of inoculant.

**Table 5 plants-14-01599-t005:** Results of soil chemical and granulometric analysis from samples collected at a depth of 0.00–0.20 m at the start of the experiment in both areas.

Location	pH	OM	P	K	Ca	Mg	H + Al	Al	SB	CEC	
	CaCl_2_	g dm^−3^	mg dm^−3^	mmol_c_ dm^−3^							
Area 1	5.8	25	3.4	0.6	21	8	16	0	29.6	45.6	
Area 2	5.2	19	3.4	1.5	25	7	19	0	31.5	50.5	
	V	m	S-SO_4_	B	Cu	Fe	Mn	Zn	sand	silt	clay
	%		g dm^−3^	mg dm^−3^					g kg^−1^		
Area 1	65	0	14	0.4	1.9	23.9	24	0.8	806	29	165
Area 2	62	0	14	0.3	1.6	35.3	29	1.2	837	42	121

P, Ca, Mg, and K: resina; S-SO_4_: Ca (H_2_PO_4_)_2_ 0.01 mol L^−1^; B: BaCl_2_.2H_2_O 0.125% microwave. Cu, Fe, Mn, and Zn; OM: organic matter; SB: sum of bases; CEC: cation exchange capacity; V: base saturation; m: aluminum saturation.

**Table 6 plants-14-01599-t006:** Soil moisture and diazotrophic bacteria in soil samples collected at the start of the field experiment in Areas 1 and 2.

Sample ID	Soil Moisture	Diazotrophic Bacteria
	%	CFU g^−1^ dry soil
Area 1	9.37	9.38 × 10^4^
Area 2	9.05	2.63 × 10^5^

CFU—Colony Forming Units.

**Table 7 plants-14-01599-t007:** Treatments with N fertilization associated with *Azospirillum brasilense* inoculation.

Treatment	Source	Dose
T1	Control	28 kg ha^−1^ N; 0 Inoculant
T2	50% N/No Inoculant	14 kg ha^−1^ de N; 0 Inoculant
T3	50% N+ Inoculant	14 kg ha^−1^ de N; 0.2 L ha^−1^ Inoculant *
T4	50% N + Inoculant	14 kg ha^−1^ de N; 0.4 L ha^−1^ Inoculant *
T5	50% N + Inoculant	14 kg ha^−1^ de N; 0.6 L ha^−1^ Inoculant *
T6	50% N + Inoculant	14 kg ha^−1^ de N; 0.8 L ha^−1^ Inoculant *

* Concentration of *Azospirillum brasilense* per mL: 2 × 10^8^ CFU mL^−1^. CFU = colony forming units.

## Data Availability

The data presented in this study are available on request from the corresponding author.

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
