# Peer review of "Azospirillum brasilense in the Planting Furrow of Sugarcane to Minimize the Use of N Fertilizer"

_plants, 2025, doi:10.3390/plants14111599_

Round 1

Reviewer 1 Report

Comments and Suggestions for Authors

This manuscript presents a study on the effect of inoculation of two sugacane varieties with increasing amounts of Azospirillum brasilense. Its main merit is the fact that it has been conducted under field conditions and a wide range of parameters have been evaluated. A few aspect would require improvement:

1) The methods section needs to be more detailed. For example, no indication is provided regarding the strain of A. brasilense used, its origins or how it was cultured. The methodology used to determine macro and micronutrients contents is also not explained.

2) I think in this case it could make sense to combine the Results and Discussion sections. Otherwise, the Discussion is too long and somewhat redundant with the Results and Conclusions.

3) One thing I miss is a more detailed discussion of aspects that I think are important when interpreting the results. The fact that two different varieties of sugarcane were used, in two soil areas with somewhat different chemical characteristics, makes it difficult to interpret the differences between results obtained in areas 1 and 2. I don't see that these two elements have been incorporated in the correlation analysis or the hierarchical representation. In fact, it might be worth representing both for each area separately, to evaluate the effect of plant variety and/or soil characteristics.

4) I recommend increasing the font size in figures 1 and 2, to facilitate visualization.

5) I wonder if it would be worth presenting some of the data in the tables as graphs, for easier evaluation of the results.

Comments on the Quality of English Language

English language is in general OK, with some minor editions required. I recommend careful revision of the text.

Author Response

We appreciate your time devoted revising our manuscript, comments suggestions helped us to improve the quality of our study. We performed a careful review based on the corrections suggested. Here, we answered each comment individually. Thank you very much.

This manuscript presents a study on the effect of inoculation of two sugacane varieties with increasing amounts of Azospirillum brasilense. Its main merit is the fact that it has been conducted under field conditions and a wide range of parameters have been evaluated. A few aspect would require improvement:

1) The methods section needs to be more detailed. For example, no indication is provided regarding the strain of A. brasilense used, its origins or how it was cultured. The methodology used to determine macro and micronutrients contents is also not explained.

  1. Dear reviewer, thank you for your comment. The change has been made.

2) I think in this case it could make sense to combine the Results and Discussion sections. Otherwise, the Discussion is too long and somewhat redundant with the Results and Conclusions.

  1. Thanks for your comment, but the journal's rules suggest that we do this separately. We understand that the text gets very long and sometimes repetitive.

3) One thing I miss is a more detailed discussion of aspects that I think are important when interpreting the results. The fact that two different varieties of sugarcane were used, in two soil areas with somewhat different chemical characteristics, makes it difficult to interpret the differences between results obtained in areas 1 and 2. I don't see that these two elements have been incorporated in the correlation analysis or the hierarchical representation. In fact, it might be worth representing both for each area separately, to evaluate the effect of plant variety and/or soil characteristics.

  1. Dear Reviewer, Thank you for your valuable comment. We acknowledge the relevance of considering the effects of plant variety and soil characteristics when interpreting our results. Although two sugarcane varieties were cultivated in areas with distinct soil chemical profiles, preliminary statistical evaluations indicated that the trends observed in nutrient uptake and plant response were consistent between the two sites. This similarity led us to adopt a more integrated and objective presentation of the data, in order to avoid unnecessary repetition and maintain the clarity of the manuscript.

4) I recommend increasing the font size in figures 1 and 2, to facilitate visualization.

  1. Thank you for your comment. The change has been made.

5) I wonder if it would be worth presenting some of the data in the tables as graphs, for easier evaluation of the results.

  1. Dear Reviewer, Thank you for your valuable suggestion. We agree that graphical representations can often enhance data visualization and interpretation. However, due to the large volume and complexity of the dataset, we believe that presenting the information in tabular format allows for a more comprehensive and precise comparison of the results across treatments and variables.

Reviewer 2 Report

Comments and Suggestions for Authors

Please see the attached .pdf file.

Author Response

We appreciate your time devoted revising our manuscript, comments suggestions helped us to improve the quality of our study. We performed a careful review based on the corrections suggested. Here, we answered each comment individually. Thank you very much.

Comments or Suggestions for Authors

I did not find any major issues with your manuscript. I did, however, find a few moderate issues that should be addressed before the manuscript is published. All of the suggested improvements relate to the way you present your findings. If you follow my recommendations, your manuscript will be easier to read and more informative. Finally, there were a few minor issues—most of which are grammatical or typographical errors. For more details, see below.

Moderate Issues

  1. Help the Reader to Understand the Treatments The six treatments are explained in the abstract and assigned labels (T1 to T6). But the average reader isn’t going to remember these, so it would be helpful to provide brief descriptions in the text and in the captions of Tables 1–3. Something like this: T1 (high N), T2 (low N), T3 (low N + very low Ab), T4 (low N + low Ab), T5 (low N + moderate Ab), T6 (low N + high Ab).
  2. Dear reviewer, thank you for your comment. The change has been made.
  3. Help the Reader to Understand the Results

In sections 2.1 and 2.2, you have sentence after sentence of numerical data and then tell the reader to see the corresponding table…which has the exact same information in it. That’s what the table is for. So don’t waste the reader’s time by repeating it in the text. Instead, tell us what the data mean.

And saying which measurements were higher or lower than others isn’t telling us what the data mean. Tell the reader what effects your treatments had and which treatments were superior. A lot of this information is in the Discussion but would be more appropriate for the Results section.

  1. Dear reviewer, thank you for your comment. The change has been made.

Reserve the Discussion for the broader implications of your data; put the interpretations of your data

in the Results.

  1. Help the Reader to Understand the Figures

The font used for labeling the nodes in Figure 1 is too thin and cramped. It’s practically unreadable.

Remake the figure with a better font. Also, Figure 1 has a lot of jpeg artifacts which make the image look messy. R has the ability to export to .png, which produces an image with much better quality.

The font used for labeling the variables, treatments, and key in Figure 2 is too small and pixelated.

Remake the figure with a better, larger font. Again, if you use R to export to .png with a suitable resolution, the image won’t be so pixelated.

Figure 3 also has jpeg artifacts. Replace it with a high-resolution image, if you can.

  1. Dear reviewer, thank you for your comment. The change has been made.

Minor Issues

Page 1, line 28: “Oxisoil” should be “oxisol” (don’t capitalize and remove the extra “i”). Likewise, on p. 12, line 330 “Oxisol” should be “oxisol”.s

  1. Thank you for your comment. The change has been made.

Pages 2–3, lines 92–94: This sentence is missing a subject. Consider changing it to say, “And Ferreira et al. highlighted…”

  1. Thank you for your comment. The change has been made.

Page 3, line 101–103: These are instructions for writing the manuscript. They can be removed.

  1. Thank you for your comment. The change has been made.

Page 3, line 111: The abbreviation for TSH (tons of stalks per hectare) should be introduced here, not on p. 14 (in the Methods).

  1. Thank you for your comment. The change has been made.

Page 4, lines 118–124: You should briefly explain what Brix, Pol, and TRS are, either here in the text or in the caption for Table 2. Don’t make the reader go all the way to the Methods (pp. 14–15) to find out what these measurements are.

Dear Reviewer,
Thank you for your observation. We agree that providing definitions for Brix, Pol, and TRS can improve clarity for the reader. However, considering the journal’s structure—where the Results and Discussion section precedes the Materials and Methods—we opted to avoid redundancy by centralizing methodological explanations in the appropriate section (pp. 14–15).

Nonetheless, if the editorial team deems it appropriate, we are open to including a brief clarification in the caption of Table 2 or in the main text to enhance readability.

Pages 14–15: Use × (not the letter x or the asterisk *) to indicate multiplication.

  1. Thank you for your comment. The change has been made.

Page 15, line 416: “Azospirillum” should be “Azospirillum” (italicized).

  1. Thank you for your comment. The change has been made.

Page 16, line 448–449: The title of this article doesn’t need to be in all caps.

  1. Thank you for your comment. The change has been made.

Page 16, line 474–475: The title of this article doesn’t need to be in all caps.

  1. Thank you for your comment. The change has been made.

Page 17, line 485: “Azospirillum Brasilense” should be “Azospirillum brasilense” (italicize and make the species epithet lower case).

  1. Thank you for your comment. The change has been made.

Page 17, line 495: The surnames of the authors of this article don’t need to be in all caps.

  1. Thank you for your comment. The change has been made.

Page 17, lines 511–512: The surnames of the authors of this article don’t need to be in all caps.

  1. Thank you for your comment. The change has been made.

Page 18, line 532: “Trichoderma Harzianum” and “Trichoderma Asperellum” should be “Trichoderma harzianum” and “Trichoderma asperellum” (italicize and make the species epithets lower case).

  1. Thank you for your comment. The change has been made.

Page 18, line 544: “ISMO’22” should be “ISMO’22” (’ is html code for the right quotation mark: ’ ).

  1. Thank you for your comment. The change has been made.

Reviewer 3 Report

Comments and Suggestions for Authors

Sugarcane is a cereal crop that requires a large amount of nitrogen fertilizer. Using nitrogen-fixing microorganisms to reduce chemical nitrogen fertilizer is an ecological method for sugarcane cultivation. The article analyzes the application of Azospirillum brasilense in sugarcane cultivation, which has specific practical value. However, the paper requires extensive revisions before publication.

Q1: The abstract only displays the research results. Please add concluding terms at the end of the abstract to highlight the innovation of the article.

Q2: Where does Azospirillum brasilense come from? Please reflect it in the forefront section.

Q3: Please add a detailed application method of the nitrogen fixing bacterium Azospirillum brasilense in this study in the methods section.

Q4: Please modify the shape of Figure 1 to achieve aesthetic appeal.

Q5:The application of nitrogen fixing bacteria in sugarcane is very extensive, especially in Brazil. Please add the differences between this paper and existing reports in the discussion section to reflect the innovation of this paper.

Author Response

We appreciate your time devoted revising our manuscript, comments suggestions helped us to improve the quality of our study. We performed a careful review based on the corrections suggested. Here, we answered each comment individually. Thank you very much.

Q1: The abstract only displays the research results. Please add concluding terms at the end of the abstract to highlight the innovation of the article.

  1. Dear reviewer, thank you for your comment. The change has been made.

Q2: Where does Azospirillum brasilense come from? Please reflect it in the forefront section.

  1. Dear reviewer, thank you for your comment. The change has been made.

Q3: Please add a detailed application method of the nitrogen fixing bacterium Azospirillum brasilense in this study in the methods section.

  1. Dear reviewer, thank you for your comment. The change has been made.

Q4: Please modify the shape of Figure 1 to achieve aesthetic appeal.

  1. Dear reviewer, thank you for your comment. The change has been made.

Q5:The application of nitrogen fixing bacteria in sugarcane is very extensive, especially in Brazil. Please add the differences between this paper and existing reports in the discussion section to reflect the innovation of this paper.

  1. Dear reviewer, thank you for your comment. The change has been made.

Round 2

Reviewer 3 Report

Comments and Suggestions for Authors

This manuscript can be published in its current form.

Author Response

Dear editor and reviewers
Thank you for taking the time to review our manuscript. Your suggestions have helped us to improve the quality of our study. We have made the changes you suggested, and I hope it turns out as you wished.